# Intragenic proviral elements support transcription of defective HIV-1 proviruses

**Jeffrey Kuniholm**[1], **Elise Armstrong**[2], **Brandy Bernabe**[3¤], **Carolyn Coote**[2], **Anna Berenson**[4], **Samantha D. Patalano**[4], **Alex Olson**[2], **Xianbao He**[2], **Nina H. Lin**[2], **Juan I. Fuxman Bass**[4], **Andrew J. Henderson**[1,2,3]*

**1** Boston University School of Medicine, Department of Microbiology, Boston, Massachusetts, United States of America, **2** Boston University School of Medicine, Department of Medicine, Section of Infectious Diseases; Boston, Massachusetts, United States of America, **3** Boston University School of Medicine Graduate Medical Sciences, Boston, Massachusetts, United States of America, **4** Boston University, Department of Biology, Boston, Massachusetts, United States of America

¤ Current address: Emory University, Laney Graduate School, Atlanta, Georgia, United States of America
* hender@bu.edu

**Data Availability Statement:** Relevant data are within the manuscript and its supporting information files.

## Abstract

HIV-1 establishes a persistent proviral reservoir by integrating into the genome of infected host cells. Current antiretroviral treatments do not target this persistent population of proviruses which include latently infected cells that upon treatment interruption can be reactivated to contribute to HIV-1 rebound. Deep sequencing of persistent HIV proviruses has revealed that greater than 90% of integrated HIV genomes are defective and unable to produce infectious virions. We hypothesized that intragenic elements in the HIV genome support transcription of aberrant HIV-1 RNAs from defective proviruses that lack long terminal repeats (LTRs). Using an intact provirus detection assay, we observed that resting CD4+ T cells and monocyte-derived macrophages (MDMs) are biased towards generating defective HIV-1 proviruses. Multiplex reverse transcription droplet digital PCR identified *env* and *nef* transcripts which lacked 5' untranslated regions (UTR) in acutely infected CD4+ T cells and MDMs indicating transcripts are generated that do not utilize the promoter within the LTR. 5'UTR-deficient *env* transcripts were also identified in a cohort of people living with HIV (PLWH) on ART, suggesting that these aberrant RNAs are produced *in vivo*. Using 5' rapid amplification of cDNA ends (RACE), we mapped the start site of these transcripts within the Env gene. This region bound several cellular transcription factors and functioned as a transcriptional regulatory element that could support transcription and translation of downstream HIV-1 RNAs. These studies provide mechanistic insights into how defective HIV-1 proviruses are persistently expressed to potentially drive inflammation in PLWH.

## Author summary

People living with HIV establish a persistent reservoir which includes latently infected cells that fuel viral rebound upon treatment interruption. However, the majority of HIV-1 genomes in these persistently infected cells are defective. Whether these defective HIV

**Funding:** This work was in part funded by NIH including R01 AI138960, R01DA055488 and DA047032 R61/R33 to A.J.H. and NIH NIGMS R35 GM128625 to J.I.F.B. J.K. was supported by the Immunology Training Program (NIAID T32 AI007309-31A1), and B.B. was supported by BU PREP, R25 GM125511. The funders had no role in study design, data collection and analysis, decision to publish or preparation of the manuscript.

**Competing interests:** The authors have declared that no competing interests exist

genomes are expressed and whether they contribute to HIV associated diseases including accelerated aging, neurodegenerative symptoms, and cardiovascular diseases are still outstanding questions. In this paper, we demonstrate that acute infection of macrophages and resting T cells is biased towards generating defective viruses which are expressed by DNA regulatory elements in the HIV genome. These studies describe an alternative mechanism for chronic expression of HIV genomes.

## Introduction

HIV-1 establishes a persistent infection by integrating into the host genome that can only be fully eradicated by the elimination of infected cells. Although antiretroviral therapy (ART) restricts viral replication and disease progression, it only slowly diminishes the persistent reservoir [1,2]. Furthermore, a subset of persistently infected cells harbor transcriptionally repressed HIV-1 proviruses which, upon reactivation, fuel the rapid rebound of HIV-1 replication and infection upon ART discontinuation. Understanding the mechanisms that establish and maintain persistent and latent HIV-1 infections are required for effective HIV cure strategies.

Persistent HIV-1 infection is also postulated to contribute to inflammation and chronic immune activation associated with systemic diseases, even in people treated with antiretroviral drugs that have undetectable HIV-1 expression [3–10]. People living with HIV-1 (PLWH) receiving ART continue to suffer from inflammatory associated co-morbidities including neurological deficits, frailty, and cardiovascular disease [8]. It has been demonstrated that innate immune sensing of HIV-1 intron containing RNAs in infected macrophages, results in induction of MAVS-dependent interferon type I and pro-inflammatory responses [11,12]. However, whether other mechanisms contribute to this persistent inflammation and immune dysfunction are poorly understood.

Studies examining the proviral landscape using next generation deep sequencing approaches show that the HIV-1 reservoir is diverse and dynamic, changing over the course of time [13–16]. Approximately 90% of HIV-1 proviruses sequenced from peripheral blood samples from antiretroviral treated PLWH have crippling mutations including large deletions of the genome that would prevent the generation of infectious HIV-1 particles [17,18]. A subset of CD4+ T cells harboring defective proviruses generate polypeptides that are presented in the context of MHC I. Cells presenting such peptides can be targeted by CD8+ cytotoxic T lymphocytes (CTLs), thereby shaping the HIV-1 provirus reservoir and potentially redirecting adaptive HIV-1 immunity [19,20]. These reports indicate that subsets of defective proviruses are expressed and influence immune cell function. Importantly, what regulates the expression of defective proviruses and how they contribute to persistent HIV-associated inflammation has not been reported.

We hypothesize that defective provirus transcription is regulated by intragenic transcriptional elements. We demonstrate that acute infection of resting CD4+ T cells and monocyte-derived-macrophages are biased towards generating defective proviruses, and despite these defects still express HIV-1 RNAs and proteins. Importantly, we detect these mRNAs in a subset of PLWH on ART. We also identify the intragenic transcriptional elements that regulate the expression of these aberrant RNAs. These studies provide mechanistic insights into how defective HIV-1 proviruses are persistently expressed to potentially drive inflammation.

## Results

### Acute infection of resting CD4+ T cells and macrophages generate defective HIV-1 proviruses

We hypothesized that acute HIV-1 infection of resting and quiescent cells biases infection towards defective proviruses. To determine the ratio of intact and defective provirus present in resting CD4+ T cells, activated CD4+ T cells, and macrophages following acute HIV-1 infection, we utilized an intact provirus detection assay (IPDA) [18] (Figs 1A and *S1A and S1 Table*). CD4+ T cells were either unstimulated or activated with anti-CD3/anti-CD28 beads prior to infection with HIV-1$_{VSVgNL4-3}$, whereas monocyte derived macrophages (MDMs) were infected with R5-tropic HIV-1$_{NL4-3\ BaL}$. Infected cells were cultured for at least 48-hours to allow the completion of reverse transcription and integration (*S1B Fig*). Genomic DNA from infected cells was used as template for IPDA droplet digital PCR (ddPCR). IPDA showed that activated CD4+ T cells had a higher frequency (>90%) of intact HIV provirus when compared to resting CD4+ T cells or MDMs (Fig 1A and 1B) which had on average 25% and 40% intact proviruses, respectively. Approximately 50–60% of proviruses detected in both resting CD4+ T cells and MDMs were defective with the majority of proviruses harboring 5' mutations that prevented detection by the *psi* probe. The non-nucleoside reverse transcriptase inhibitor Efavirenz (EFV) was used as a negative control to assure that IPDA signals were dependent on reverse transcription and were not contaminating DNA (*S2 Fig*). These results support that defective HIV proviruses are generated in both resting CD4+ T cells and MDMs after an acute infection, and the activation state of CD4+ T cells influences efficient establishment of intact proviruses.

### Defective proviruses generate proviral RNAs that lack 5'untranslated regions

Several studies demonstrated that defective HIV proviruses are transcribed and translated [19,20]. To determine if the provirus populations detected in HIV-1 infected CD4+ T cells and

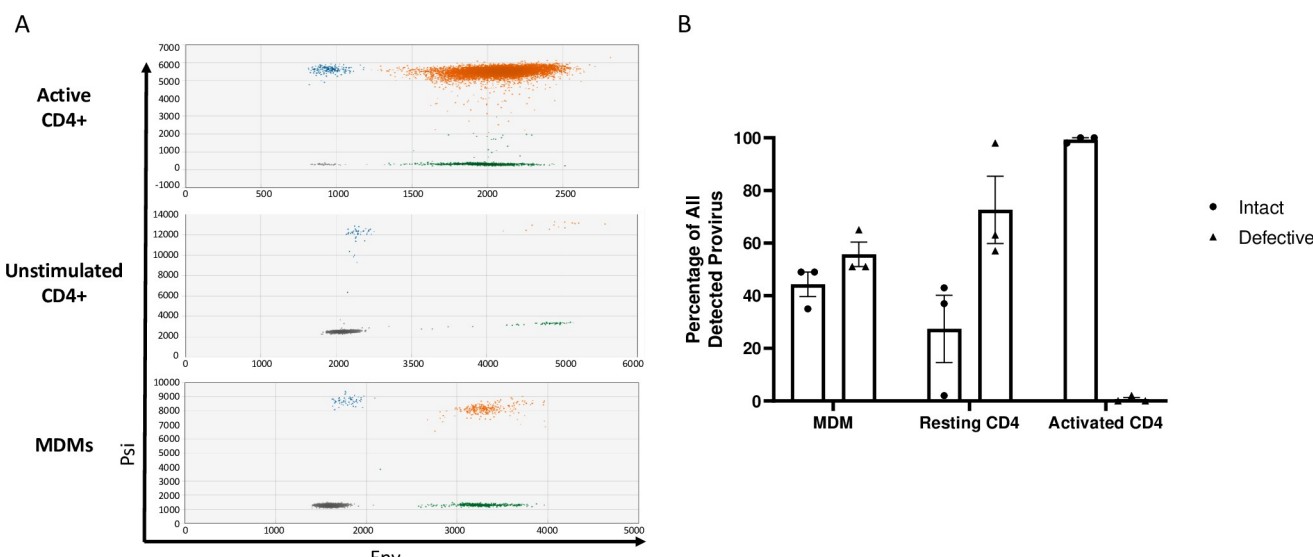

**Fig 1. Acute HIV-1 infection generates defective proviruses in MDMs and resting CD4+ T cells.** (A) Representative raw IPDA data of (top) activated CD4+ T cells, (middle) unstimulated CD4+ T cells, and (bottom) monocyte-derived macrophages (MDMs) infected with HIV-1. Droplets are color coded based on manual gating of positive and negative probe signals (gray = empty or double mutation/deletion droplets, blue = droplets single positive for *psi*, green = single positive *env* droplets, orange = double positive *psi* and *env* intact proviral droplets). A parallel reaction to detect the host cell gene *RPP30* was used as a correction for DNA shearing. (B) Percentages of intact and defective HIV genomes detected in MDMs and CD4+ T cells following HIV-1 infection. MDMs were infected with HIV-1$_{NL4-3-BaL}$ for 48 hours prior to DNA isolation. Resting and anti-CD3/CD-28 bead activated CD4+ T cells were infected with HIV-1$_{NL4-3-VSVg}$ for 72 hours prior to DNA isolation. Data represents three independent infections using cells generated from three different donors. Resting and activated CD4+ T cell sample data are participant matched.

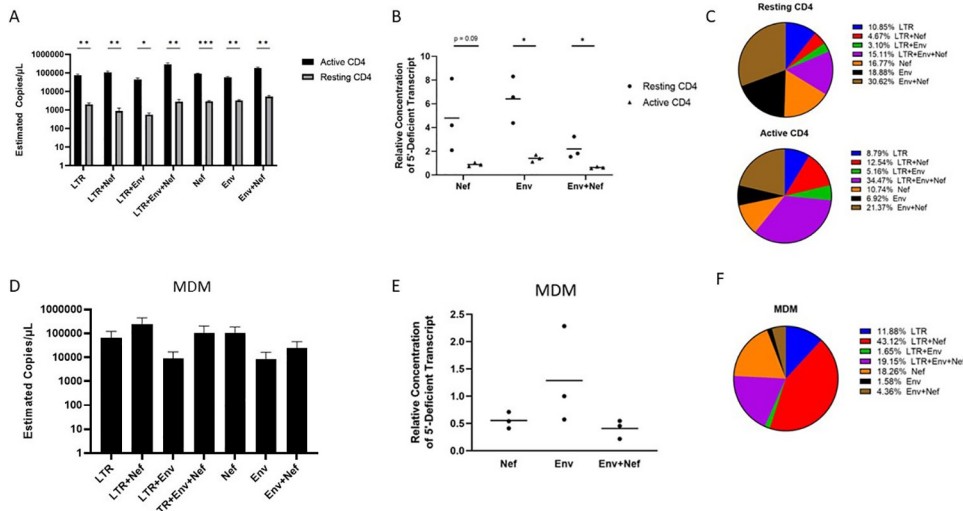

**Fig 2. CD4+ T cells and MDMs express aberrant HIV-1 RNAs.** (A) Multiplex RT-ddPCR was performed on RNA prepared from unstimulated and activated CD4+ T cells infected with HIV-1$_{NL4-3}$ by spinoculation for 90 minutes at an MOI of 0.02 and incubated for 72 hours prior to RNA isolation. HIV-1 infection was limited to a single round by addition of 10 μM saquinavir 30 minutes after spinoculation. Activated CD4+ T cells were cultured with anti-CD3/CD28 beads for 72 hours prior to spinoculation. Estimated copies/μL for transcripts were calculated with the QuantaSoft droplet reader software after manual gating of multiplex RT-ddPCR data (See S3 Fig) and dilution factor correction. (B) Ratios of HIV 5'UTR defective transcripts relative to transcripts with intact 5'UTRs for resting and activated HIV infected CD4+ T cells. (C) Frequencies of detected HIV transcripts as a percentage of all transcripts quantified by multiplex RT-ddPCR for resting (top) and CD3/CD28 activated CD4+ T cells. (D) Multiplex RT-ddPCR performed with RNA from HIV-1$_{NL43-BaL}$ infected MDMs. MDMs were infected with HIV-1$_{NL43-BaL}$ at an MOI of 0.01 for 4 hours by adding virus directly to adherent cell cultures for 48 hours prior to RNA isolation. (E) Ratios of 5'UTR defective transcripts relative to transcripts with intact 5'UTR for HIV infected MDMs. (F) Frequencies of HIV-1 transcripts as a percentage of all transcripts quantified by multiplex RT-ddPCR for HIV infected MDMs. Data represent three independent experiments using cells from three donors. Resting and activated CD4+ T cell sample data are donor matched. Bars represent standard error of the mean. Significance values were generated using multiple unpaired Two-tailed T tests. * denotes p < 0.05; ** denotes p < 0.01, and *** denotes p<0.001.

MDMs were transcriptionally competent, we adapted a reverse transcription ddPCR assay (RT-ddPCR) originally described by Yukl et al [21] by multiplexing probes for simultaneous detection of transcripts spanning the R-U5-Gag junction ("Late LTR" probe), distal transcripts ("Nef" probe), and sequences that included both upstream and distal sequences (5' UTR/Nef double positives, S3 Fig) [22,23]. Multiplex RT-ddPCR performed using infected CD4+ T cell cDNA detected transcripts which included 5' untranslated region (UTR) sequence (R-U5-gag positive) only and double-positive 5' UTR-Nef transcripts. We also detected *nef*-positive transcripts which did not include 5' UTR sequence in HIV infected CD4+ T cells suggesting these transcripts were not generated from the transcriptional start site located in the 5' LTR (Fig 2A). We further multiplexed the RT-ddPCR by including an *env* primer and probe combination allowing for the detection of 7 different HIV-1 RNAs (S3 Fig). The multiplex RT-ddPCR identified two general sets of transcripts in CD4+ T cells and MDMs; a set of RNAs that contained 5' UTR sequences and a set of RNAs that were positive for distal RNAs *env* and *nef* but lacked 5' UTRs. The relative levels of RNAs with and without 5' R-U5 sequence differed between the infected CD4+ T cell populations with approximately 40% of RNAs in activated CD4+ T cells lacking 5' UTRs compared to 66% of RNAs lacking 5' UTRs in resting cells. These frequencies correlated with IPDA data which showed that resting CD4+ T cells harbor more defective proviruses (Fig 2B and 2C). We also observed approximately 25% of RNAs in infected MDMs lacked 5' UTRs (Fig 2D–2F). Using a poly d(T) primer instead of random

hexamers for the RT reactions resulted in the amplification and detection of 5' UTR deficient transcripts in CD4+ T cells and MDMs, indicating that a subset of these RNAs were polyadenylated (*S4 Fig*). These data support that aberrant HIV transcripts are produced after acute HIV-1 infection, with their relative expression influenced by cell type and activation state.

Chronically infected individuals on ART demonstrate a gradual decrease in intact HIV proviruses concurrent with selection and expansion of defective proviruses due to sustained immunological pressure and clearance [15,17–19,24]. Notably, defective HIV proviruses are observed to decay at a slower rate than intact provirus clones, presumably due to their relatively low HIV-1 expression levels [13,23–26]. We posit that HIV-1 mRNA lacking 5' UTRs should be detected in people living with HIV (PLWH) on ART. Using DNA isolated from PBMCs of PLWH on ART (*S2 Table*), we first confirmed the presence of HIV-1 proviruses in chronically infected individuals by IPDA with most proviruses detected in these individuals being defective (Fig 3A). To determine if aberrant RNAs that lack 5'-UTRs were also present in individuals chronically infected with HIV-1, multiplex RT-ddPCR was performed using RNA from PLWH donor PBMCs. Probes specific for R-U5 5' UTR sequences and *env* were used in tandem to determine the presence of one or both probe sites on HIV-1 transcripts expressed in HIV-positive donor PBMCs. Multiplex RT-ddPCR assays successfully detected HIV-1 RNAs with 5' UTR as well as *env* transcripts that lacked 5' UTRs start sites (Fig 3B and 3C). *Env* sequences lacking 5' UTRs were detected with signal above uninfected PBMC background controls in 10/11 RNA samples from HIV-positive PBMC donors (Fig 3B and *S2 Table*). These data show that aberrant HIV transcripts are detected in a subset of individuals chronically infected with HIV even during ART suggesting these RNAs are persistently expressed in PLWH.

## Defective proviral transcripts are generated from intragenic transcriptional elements

In the context of intact proviruses, the 5' LTR includes the cis-regulatory elements that recruit and coordinate host transcription factors to initiate and regulate HIV-1 transcription. The mechanisms by which HIV transcription is regulated in defective HIV proviruses that lack a functional 5' LTR remains incompletely described. We hypothesized that there are intragenic sequences that support transcription of defective HIV-1 proviruses. Previous reports that support this hypothesis include descriptions of transcription factor binding and DNase I hypersensitive sites downstream of the HIV-1 LTR [27–32].

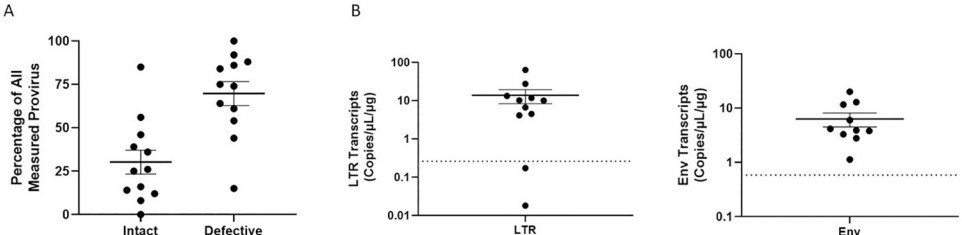

**Fig 3. Detection of HIV-1 aberrant transcripts in PLWH on ART.** (A) Percentage of intact and defective HIV provirus detected in DNA samples from PBMCs of HIV-positive ART-treated individuals relative to all provirus quantified by IPDA. (B) Multiplex RT-ddPCR was performed to detect intact 5' UTR containing transcripts (LTR) and 5' UTR-deficient *env* transcripts (Env) using RNA samples prepared from PBMCs of HIV-positive ART-treated individuals. No RT enzyme control reactions were carried out for each assay and background signal was subtracted from experimental readings. Dots represent estimated copies/uL of single positive transcripts per μg of RNA. Dotted line represents the mean assay background signal calculated from five HIV-negative PBMC samples. Each point represents an independent participant sample. A total of 11 participant samples were assessed.

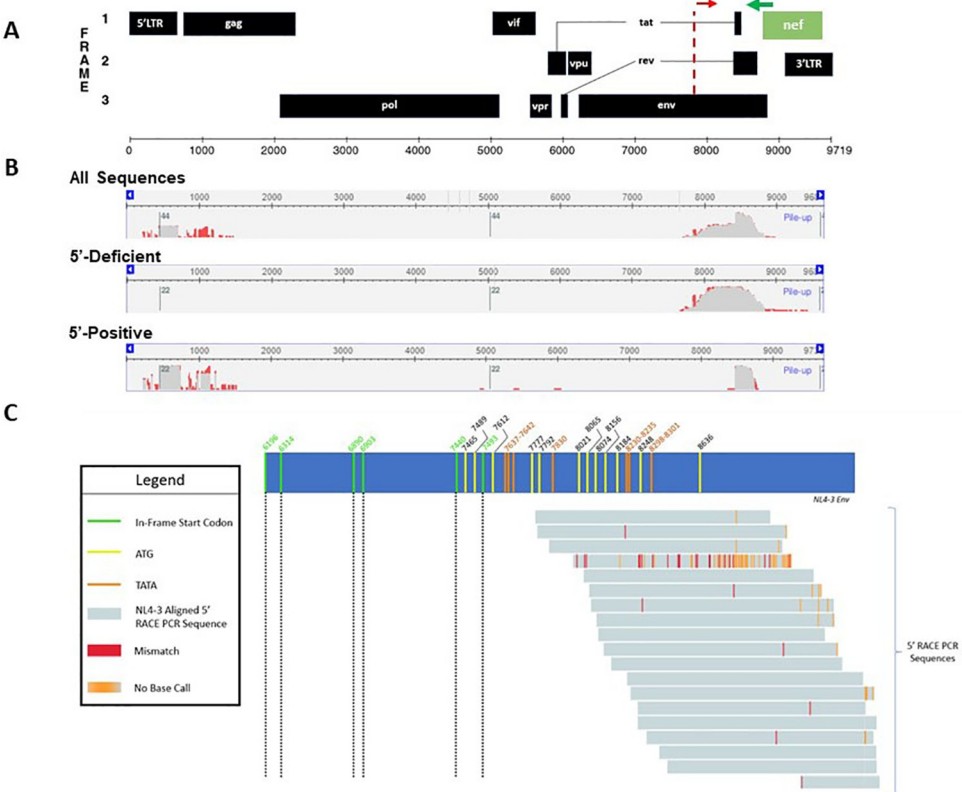

**Fig 4. Sequencing of 5' RACE PCR products identified HIV-1 transcripts with intragenic start sites.** (A) Schematic of HIV-1 genome with putative promoters indicated as dotted red lines as predicated by Promoter 2.0 [33] and ElementNT V2 [34]. The green line represents the 5' RACE gene-specific Nef primer. (B) 5' RACE results analyzed using the NCBI Multiple Sequence Alignment Viewer. (Top) All 5' RACE sequences aligned to NL4-3 sequence (N = 41 sequences). (Middle) 5' RACE sequences which lacked 5' UTR aligned to NL4-3 sequence (N = 19). (Bottom) 5' RACE sequences which contained 5' UTR aligned to NL4-3 sequence (N = 22). (C) Schematic alignment of 5' RACE sequences lacking 5' UTR sequence (gray bars, N = 19) with the NL4-3 env gene (blue bar). Green and black dotted lines indicate in-frame start codons (numbered using GenBank: AF324493.2). Yellow lines represent out of frame start codons. Orange lines (top) represent TATA box sequences. Red lines represent sequence mismatches between 5' RACE reads and NL4-3 *env*. Orange lines (bottom) indicate that no nucleotide was assigned for that position during the sequencing reaction. Start codons and TATA box sequences were mapped and multiple sequence alignments were performed using Benchling Biology Software (https://benchling.com, 2021).

To characterize transcriptional start sites of defective HIV transcripts, we performed 5' RACE PCR using RNA isolated from HIV infected activated CD4+ T cells and MDMs. A universal forward primer and a gene-specific reverse primer for *nef* (*S1 Table*) were used to amplify the populations of *nef* containing transcripts from infected cells. PCR products were separated by gel electrophoresis and ~1kb RACE PCR products were gel extracted, cloned into a pUC19-based vector, and 41 products were sequenced to map their 5' ends (*S3 Table*). MDM and CD4+ T cell sequences demonstrated that 53.6% of the transcripts were splice products which initiated at the transcriptional start site within the 5' LTR sequence (Fig 4A and 4B). The remaining 46.3% of sequences initiated transcription within *env*, across a span of approximately 200 nucleotides located between +7760 to +8178 bp downstream of the canonical 5' LTR transcriptional start site (Fig 4C and *S3 Table*). These data support that transcription initiates from intragenic sites of the HIV genome. Based on the RACE data, we used Promoter 2.0 [33] and ElementNT V2 [34] to predict if promoter characteristics were within the *env* region. These bioinformatic tools predicted a transcriptional start site (7800 bp, score 1.141) and

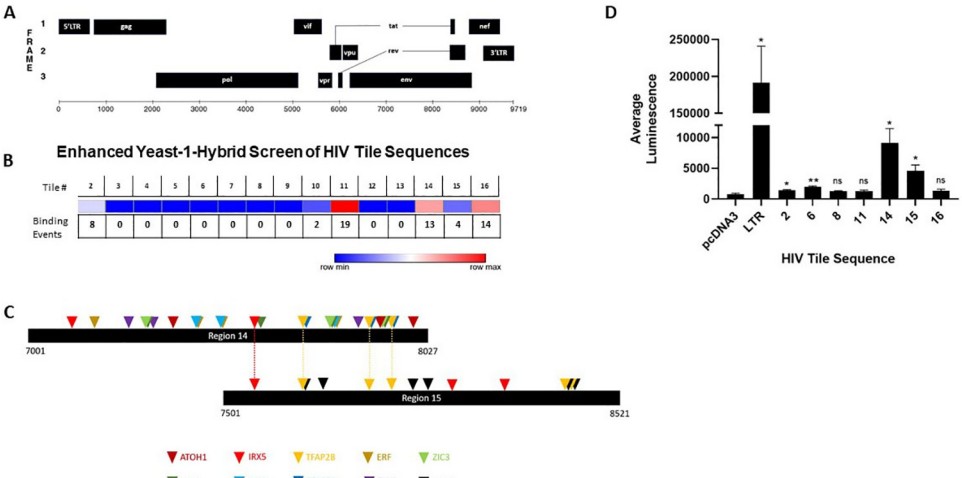

**Fig 5. Identification of intragenic transcriptional elements.** (A) Schematic of HIV-1 genome. (B) Heat map for transcription factor binding within the HIV-1 provirus. HIV-1 tile sequences (1–16) were used in yeast-1-hybrid screen (See S4 Table). Dark blue represents regions lacking transcription factor binding whereas red indicates regions of transcription factor binding. Binding events represent the number of transcription factors that bound the specific HIV-1 genomic sequence. (C) Location of predicted binding sites for transcription factors identified by yeast-1 hybrid assay that bind baits 14 and 15. Motifs were identified using FIMO with a threshold set at $p < 0.001$ ([38] and S6 Table). (D) Intragenic sequences enhance transcription. HIV-1 tile sequences were cloned upstream of a luciferase gene in pcDNA3.1 Luciferase and transfected in HEK-293T cells. Luciferase was measured and reported as fold-change in signal over cells transfected with the negative control pcDNA3.1-Luc plasmid. HIV-LTR-Luciferase was used as a positive control. These data include three independent transfections performed in triplicate. Error bars represent standard error of the mean for all values. Two-tailed unpaired T-tests were used to calculate statistical significance between raw luminescence values for HIV-1 tile sequences and negative control pcDNA3.1-Luc plasmid. * denotes $p < 0.05$; ** denotes $p < 0.01$, and *** denotes $p < 0.001$.

TATA box motifs (7483–7488 bp and 7662–7669 bp) within the *env* sequence, adjacent to the start sites identified by 5' RACE (Fig 4B and 4C). Alignment of 5' RACE PCR sequences to the NL4-3 *env* gene revealed a cluster of out-of-frame start codons in this region, suggesting that multiple alternative transcriptional start sites may be operative in the context of defective provirus (Fig 4C). To survey direct binding of human transcription factors within the HIV proviral genome, we performed a functional enhanced yeast one-hybrid screen (eY1H) [35–37]. This method allows for high-throughput screening of host transcription factor binding events to the HIV-1 genome. In brief, HIV bait sequences were generated by tiling the HIV-1$_{NL4-3}$ genome into 17 overlapping fragments (S4 Table). Fragments were cloned upstream of the two reporter genes, *LacZ* or *HIS3*. Yeasts were transformed with reporter constructs as well as a library representing 66% of the known human transcription factors fused to the yeast activation domain, *Gal4*. Transcription factor binding hits were identified as yeast colonies that grew in the absence of histidine and were blue when provided X-Gal. The eY1H screen detected 60 transcription factor-DNA interactions, representing 43 unique transcription factors which bound to six regions within the HIV-1 genome (designated 2, 10, 11, 14, 15 and 16) (S5 Table and Fig 5A and 5B). Transcription factor binding hot spots were clustured within and near HIV *env* gene (Fig 5A–5C), whereas, transcription factor binding was not detected in *pol* (Regions 3–9). Identified transcription factors included homeobox, basic helix-loop-helix, and zinc finger families (S5 Table). The distribution of transcription factor binding hotspots near the HIV *env* gene correlated with transcriptional start sites suggested by the 5' RACE results. To investigate if these intragenic sequences that bound host transcription factors function as cis-regulatory elements, reporter vectors were generated with HIV-1 fragments cloned

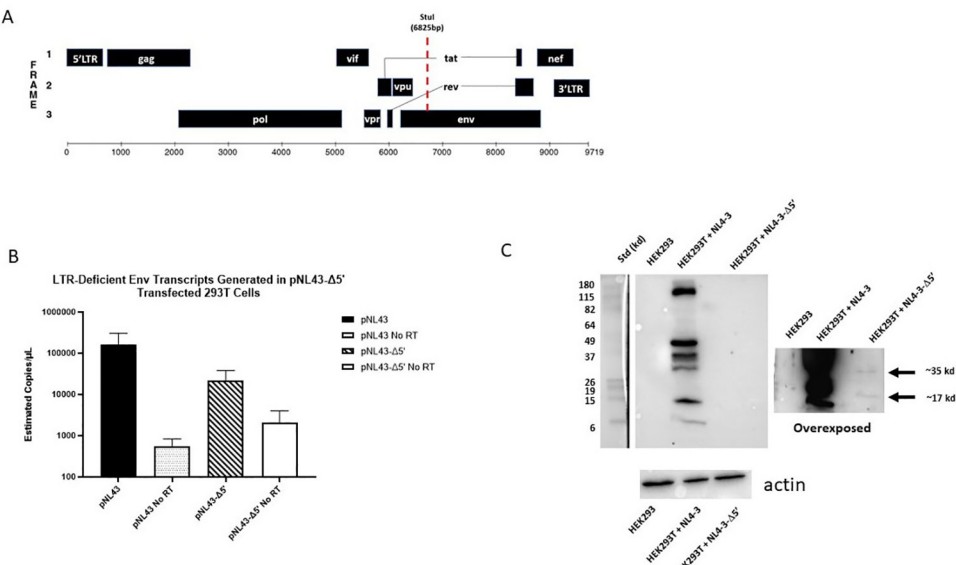

**Fig 6. Truncated HIV-1 proviral construct expresses RNA and protein.** (A) Schematic of HIV-1 NL4-3 in pNL4-3 expression plasmid and location of *StuI*-restriction digest site used to generate pNL4-3Δ5' (bottom). (B) RT-ddPCR data from HEK-293T cells transfected with either full length pNL4-3 (black) or pNL4-3Δ5' (striped) for 24-hours. RNA samples were subjected to multiplex ddPCR using a probe for the 5'LTR and a probe for env and droplets were quantified as indicated above. Reaction mixtures without RT enzyme (No RT) were used as negative controls for contaminating plasmid DNA. No LTR signal was detected for HEK293T cells transfected with pNL4-3Δ5'. Data include 4 independent pNL4-3 transfections and three pNL4-3 Δ5' transfections. (C) Western blots using cell lysate from HEK293T cells transfected with either full length pNL4-3 plasmid or pNL4-3Δ5'. Lysates were run on 12% SDS-PAGE gel, transferred to a PDVF membrane, and probed with human polyclonal anti-HIV IgG. Two exposures are shown: a short exposure demonstrating HIV-1 proteins (left) and an extended exposure which shows weaker HIV protein bands at approximately 35 kDa and 19 kDa, as indicated by the arrows (right). Non-transfected HEK293T cell lysates were also probed as a negative control. The lower panel is a separate gel from the above lysates probed with a monoclonal against β-actin antibody demonstrating the presence of proteins in the different lysates.

upstream of the luciferase reporter gene. Reporter plasmids were transiently transfected in HEK293T cells, and firefly luciferase was assessed 24 hours post-transfection (Fig 5D). Most of the intragenic sequences, including Regions 11 and 16, which bound several transcription factors in the eY1H assays, modestly influence transcription of the reporter gene. However, regions 14 and 15, which spanned sequences 7001 to 8521 bp in *env* and bound several transcription factors in the eY1H assays, exhibited 5–10 fold higher luciferase signal compared to pcDNA3 control. These sequences also spanned sites that included putative promoter elements and the RACE transcriptional start sites. The predicted transcription factor binding motifs generated using Finding Individual Motif Occurance (FIMO) software [38] for the sequences that span regions #14 and #15 are shown in *S6 Table*.

In addition, we generated a viral expression construct, pNL4-3Δ5' in which we deleted approximately 6,000 bp of the 5' end of the provirus, including the 5' LTR, *gag* and *pol* genes, and transfected this construct into HEK293T cells (Fig 6A). Transcription of *env* containing sequences was detected by RT-ddPCR a log-fold higher than the assay background (Fig 6B). Furthermore, 35 kd and 19 kd proteins were detected by immunoblotting when whole cell lysates from pNL4-3Δ5' transfected cells were probed with human HIV anti-serum suggesting that these RNAs are translated (Fig 6C). It should be noted that immunoblots required longer exposure times to visualize the proteins expressed from pNL4-3Δ5', probably reflecting that these intragenic elements are much weaker than the HIV-1 5'-LTR which increased luciferase expression on the order of ~200-fold compared to pcDNA control in transfected HEK293T

cells (Fig 5D). Taken together, these data support that HIV-1 intragenic sequences, including those within the *env* gene, act as cis-regulatory elements which support the transcription of cryptic RNAs from a subset of defective proviruses.

## Discussion

The reservoir of persistent HIV-1 proviruses is dynamic, being shaped over time by immune recognition and clearance of cells expressing HIV-1 proteins [17,18,39], as well as clonal expansion and homeostasis of memory T cell subsets that harbor HIV-1 proviruses [13,14,17]. This model is supported by longitudinal tracking of the persistent HIV-1 proviruses in PLWH on ART and long-term controllers which demonstrated a relatively rapid decay of intact HIV-1 provirus compared to defective proviruses [23–25,40]. These dynamics result in the progressive accumulation of defective HIV-1 proviral genomes which comprise the vast majority of the latent reservoir in chronically infected individuals that are undergoing treatment or are controllers of HIV infection [13–16,23,24,40]. In this study, we provide evidence that a subset of these defective proviruses are expressed during acute HIV-1 infection and provide insights into the mechanisms that control their transcription.

We hypothesized that certain cells may be predisposed to generating defective proviruses upon infection. It has been observed that quiescent and resting cell populations such as macrophages, resting CD4+ cells and memory T cell subsets are difficult to infect and biased towards unproductive or latent infections [41–43]. We compared the fate of HIV-1 infections in macrophages, unstimulated CD4+ T cells, and CD4+ T cells activated through the CD3/CD28 signaling axis. At the time of infection, macrophages and unstimulated CD4+ T cells harbor more defective proviruses compared to activated CD4+ T cells with up to 80% of the infected cells containing 5' deletions or mutations. Our results suggest that upon acute infection, even in the absence of immune selection, a foundation of cells harboring defective viruses is rapidly established [44].

The shaping of the reservoir and persistent proviral genomes has been linked to provirus expression and selection by anti-HIV immune responses. For example, it has been shown that a subset of defective viruses from individuals treated with ART can be expressed and targeted by CD8+ T cells [17,39]. However, it is not clear how these defective proviruses are transcribed, especially if a significant number of proviruses have large upstream deletions or LTR mutations. The existence of intragenic promoters has been suggested for several retroviruses [45–49]. The presence of proviral cis-regulatory elements has also been proposed within the HIV-1 genome, with reports of DNAse hypersensitive sites, methylated CpG islands, transcription factor DNA binding sites and modest transcriptional activation potential [27,31,32,50,51]. However, the regulatory mechanisms which control transcription of alternative transcriptional start sites in HIV remain largely undescribed. We hypothesized that intragenic cis-elements exist and drive transcription of a subset of the defective HIV proviral genomes. Taken together, our data generated from yeast-one-hybrid transcription factor binding assays, promoter prediction software, transfections with deletion and reporter plasmids, and 5' RACE sequencing, indicate cis-transcriptional elements and promoters located at approximately +7,800 bp in the HIV-1 proviral genome. This observed transcriptional activity overlaps with previously reported CpG islands [51]. The transcripts generated originate in the *env* gene, span the *tat* splice junction, and include intronic sequence.

Despite minimal detectable virus replication in PLWH undergoing anti-retroviral treatment, immune stimulation and inflammation persist, contributing to co-morbidities including neuroinflammation, cardiovascular disease, and signs of accelerated aging [52–54]. Previous work has indicated that HIV-1 transcripts are produced by defective proviral clones and a

subset of these transcripts can be translated into viral proteins that stimulate CD8+ T cell activity [19,20,39]. Furthermore, cryptic HIV-1 peptides are produced by using alternative reading frames (ARFs) which are distributed throughout the HIV-1 genome. Previous studies have demonstrated that ARFs can be loaded into and presented by MHC-I to activate CD8+ T cells, inducing cytokine release and killing activity [55–57]. Taken together, these data support that defective persistent HIV proviruses influence and perpetuate immune responses against HIV. Our ability to detect HIV-1 protein from proviral sequences that lack intact 5' sequences suggest that defective viruses may provide and alternative mechanism to generate ARFs.

In addition, it has been demonstrated that partially spliced and unspliced HIV-1 RNAs are recognized by MAVS-dependent nucleic acid sensing pathways and mediate IFN-I responses [11,12,58]. The defective mRNA sequences generated from the intragenic elements that we have characterized initiate transcription within *env* and span the intron leading us to speculate that these mRNAs could be proinflammatory. Importantly, we detect transcripts that are initiated from intragenic elements in samples from PLWH on ART. Overall, our study provides potential insight into the mechanisms and proviral elements which regulate transcription of these defective HIV-1 genomes and the potential of cryptic peptides that perpetuate HIV pathogenesis.

## Methods

### Ethics statement

These studies utilize PBMCs from PLWH treated with ART enrolled in a study assessing inflammation and immune dysfunction in people living with HIV and aging which was approved by Boston University IRB (IRB-33095) [10]. Inclusion criteria included individuals between 18–35 and $\geq$50 years old who were on antiretroviral treatment for at least 6 months. Subjects with recent active infection (30 days) or on immunosuppressive therapy were excluded from enrollment. Written informed consent was obtained prior to specimen and data collection. See S2 Table for details. PBMCs were isolated using Ficoll-Hypaque density gradient separation and cryopreserved at -150°C until use.

### Cells

HEK293T cells (ATCC) were maintained and cultured in Dulbecco's Modified Eagle Medium (DMEM; Invitrogen) supplemented with 100 U/mL penicillin/streptomycin (P/S; Invitrogen), 2 mM L-glutamine (Invitrogen), and 10% fetal bovine serum (FBS) (Gemini Bio-Products). CD4+ T cells were enriched from peripheral blood mononuclear cells (PBMC) from leukapheresis packs (New York Biologics). Negative selection was performed using EasySep Human CD4+ T Cell Enrichment Kit (Stemcell Technologies) after Lymphoprep density gradient (Stemcell Technologies) separation of PBMCs. Resting CD4+ T cells were cultured in Roswell Park Memorial Institute medium (RPMI; Invitrogen) with 100 U/mL P/S, 2 mM of L-glutamine, and 10% FBS. Flow cytometry using CD25 and CD69 markers confirmed resting CD4 + T cell phenotypes [42]. CD4+ T cells were activated with anti-CD3/CD28 Dynabeads (Invitrogen) at a ratio of one bead per cell for 72 h and maintained in RPMI with 100 U/mL IL-2 (AIDS Reagents Program) and 100 ng/mL IL-7 (Miltenyi Biotec). Monocyte-derived-macrophages (MDMs) were generated by positive selection of CD14+ cells from PBMCs using the EasySep Human CD14 Positive Selection Kit II (STEMCELL Technologies, Cat: 17858) and differentiating cells for at least 7 days in RPMI supplemented with 10% Human AB Serum (Millipore Sigma), 100U/mL P/S, 2mM of L-glutamine, and 20 ng/mL M-CSF (BioLegend, Cat: 574802) at 37°C and 5% $CO_2$. 50% fresh media was added 2 days after plating followed by full media changes every 2–3 days to remove non-adherent cells.

## Viruses and infections

Viral stocks of molecular clones HIV-1$_{NL4-3}$ and HIV-1$_{NL4-3\ BaL}$ (AIDS Reagents Program) were generated by transfecting HEK293T cells using PEI transfection reagent (Sigma, Cat: 408727). Pseudotyped HIV-1$_{VSVg-NL4-3}$, HEK293T cells were co-transfected with an expression vector containing VSVg and HIV-1$_{NL4-3}$. HIV-1$_{NL4-3}$ virus stocks titers were estimated using CEM-GFP cells obtained from the AIDS Reagent Program [59]. HIV-1$_{NL4-3\ BaL}$ virus stocks were titrated on primary MDMs and IU/mL were estimated from IPDA results. All infections were performed using MOI < 1 (HIV-1$_{VSVg-NL4-3}$ MOI = 0.1, HIV-1$_{NL4-3}$ MOI = 0.02, HIV-1$_{NL4-3\ BaL}$ MOI = 0.01). DNAse/MgCl2+ was added to cells at time of infection prior to DNA/RNA isolation to limit contaminating plasmid DNA. CD4+ T cells were infected by spinoculation as previously described [60]. MDMs were infected by adding HIV-1$_{NL4-3BaL}$ directly to the cells at 37˚C for 4 h, washed twice with PBS and maintained in fresh media. 1μM of the non-nucleoside reverse transcriptase (RT) inhibitor Efavirenz was added prior to infection and post-infection as a negative control.

## DNA/RNA isolation

DNA and RNA were isolated from cells using the AllPrep DNA/RNA Mini Kit (Qiagen, Cat: 80204). Briefly, cells were washed twice with PBS, treated with DNAse for 10 min prior to lysis directly in cell culture plate (MDM) or as cell pellets (CD4+ T cells) using Qiagen RLT buffer supplemented with β-Mercaptoethanol. Cell lysates were homogenized with Qiashredder columns (Qiagen, Cat: 79654) following manufacturer's protocol for AllPrep DNA/RNA Mini Kits. RNA isolation columns were treated with DNAse for > 10 minutes to remove contaminating DNA.

## Intact Proviral DNA Assay (IPDA)

Probes targeting 5' and 3' regions of the HIV provirus were designed to bind minimally variant locations as described in detailed by Bruner et al [18]. Briefly, primers and probes to amplify signal from the psi region and RRE of the env region of the HIV-1 provirus simultaneously were used (S1 Table). Signals were identified using a Bio-Rad QX200 Droplet Digital PCR System and Quantasoft Data Analysis Software.

A hydrolysable FAM probe was designed to bind Psi sequence whereas hydrolysable VIC probe was designed to bind the RRE sequence. Primers flanked probes. An unlabeled "dark probe" specific for a commonly hypermutated env sequence was included in each IPDA reaction to distinguish hypermutated provirus sequences. Proviruses containing hypermutated signals were detected as single-positive droplets containing FAM-only signal.

## Droplet Digital PCR (ddPCR)

Droplet digital PCR Reaction mixes were comprised of: 1X ddPCR Supermix (no dUTP) (Bio-Rad, Cat: 186–3024), 750 nM primers, 250 nM probes. DNA was diluted to ~10–100 ng in 8 μL of nuclease free H$_2$O per reaction. A control reaction using probes and primers for human RPP30 gene was performed to calculate shearing index and correct the estimated concentrations of intact and defective HIV-1 proviruses. Droplets were generated using Bio-Rad's QX200 Droplet Generator (Cat: 1864002).

PCR conditions were: 95˚C for 10 min enzyme activation, 94˚C for 30 sec denaturation, 59˚C for 1 min annealing/extension, repeat denaturation and annealing/extension 45 cycles, 98˚C for 10 min to inactivate enzyme.

Droplets containing PCR product were assessed on the Bio-Rad QX200 Droplet Digital PCR System. Data were collected using Quantasoft Software set to copy-number variation (CNV) mode. All experiments included a no template (water only) control and/or an uninfected cell sample negative control for background probe signal. RPP30 single probe was used to estimate shearing index and to correct the estimation of defective and intact proviruses. Estimations of intact and defective provirus frequencies were calculated using droplet counts of single- and double-positives determined by the Quantasoft software.

For RT-ddPCR, RNA was reverse transcribed to generate cDNA with RT reaction mixtures of 14 µl of 5× SuperScript III buffer, 7 µl of 50 mM MgCl2, 3.5 µl of random hexamers (50 ng/µl), 1.35 µl of oligo dT15 (500ug/mL), 3.5 µl of 10 mM deoxynucleoside triphosphates (dNTPs), 1.75 µl of RNAseOUT (40 U/µl; Invitrogen), 2 µl of SuperScript III RT (200 U/µl; Invitrogen), and nuclease-free dH$_2$O to bring the final reaction volume to 70 µL. RT was performed at 25˚C for 10 min, 50˚C for 50 min, followed by an inactivation step at 85˚C for 5 min. Negative controls included reactions with no RT enzyme. *S1 Table* shows primers and probes for ddPCR. FAM probes were designed and directed to sites throughout expected HIV transcripts. RT-ddPCR probe design was based on those reported by Yukl et al [21]. ddPCR probes were multiplexed to measure the frequency of probe sites simultaneously in ddPCR assays.

## 5' RACE PCR

5'-RACE PCR was performed on total RNA from VSVg-NL43 infected CD4+ T cells and MDMs following the manufacturer's protocol for SMARTer RACE 5'/3' Kit (Takara Bio, Cat: 634858). Random primers were annealed to template RNA using 10X Random Primer Mix, RNA (1µg), and water and incubating the mixture at 72˚C for 3 min followed by cooling to 42˚C for 2 min. SMARTer II A Oligonucleotide (1µL) was added to tag the 5'-end of the generated cDNA providing a universal forward primer binding site for subsequent PCRs. The RT reaction included template RNA annealed with random primers, 5X First-Strand Buffer, DTT (100 mM), dNTPs (20 mM), RNase Inhibitor (40 U/µl), Reverse Transcriptase (100U) and was performed at 42˚C for 90 min and then cooled to 70˚C for 10 min. This first-strand reaction product was diluted with 90 µL of Tricine-EDTA Buffer before RACE-PCR. Sequences generated are shown in *S3 Table*.

For 5' RACE PCR a nef specific reverse primer was designed (*See S1 Table*). The RACE reaction mixture consisted of 5' RACE PCR sample: PCR-Grade H$_2$O (15.5 µL), 2X SeqAmp Buffer (25 µL), SeqAmp DNA Polymerase (1 µL), 5' RACE-Ready cDNA (2.5 µL), 10X Universal Primer Mix (5 µL), 5' gene-specific Primer (10 µM). Negative controls included only the universal primer mix or only the gene-specific primers. 5' RACE PCR thermal cycling was: 94˚C for 30 sec, 68˚C for 30 sec, 72˚C for 3 min, repeated for 25 cycles. 5' RACE-PCR products were separated by gel electrophoresis (1% agarose) and major PCR products were excised, pooled, and subcloned into a linearized puC19-based vector (pRACE) using linearized pRACE vector (1µL), Gel-purified RACE product (7 µL), and In-Fusion HD Master Mix (2 µL). The ligation reaction was incubated for 15 min at 50˚C before transforming Stellar Competent Cells (TakaraBio). Colonies were selected for direct bacterial colony sequencing using M13F/R primers which flanked the In-Fusion cloning site of the pRACE vector (Genewiz).

## Enhanced yeast-1-hybrid screen

Enhanced yeast one-hybrid (eY1H) assays were performed as previously described [35–37]. Briefly, the HIV-1 NL4-3 genome was tiled into 17 overlapping fragments of 1000-1200bp in length (*S3 Table*). "Bait" HIV DNA were cloned upstream to the yeast selection marker *HIS3*

or the reporter gene *LacZ*. Constructs were transformed and integrated into yeast chromatin. A library of 1,086 human "prey" transcription factors were linked with the yeast Gal4 activation domain. Binding of the transcription factors to the DNA fragment resulted in transcription of reporter genes. Positive interactions were determined as two of the four replicate colonies that survived in the absence of histidine and experienced a color change in the presence of the substrate X-Gal. More than 90% of positive interactions were identified in 4/4 colonies [35–37]. *S5 Table* highlights transcription factors that bound HIV-1 tile sequences. Putative binding sites were predicted by FIMO software [38] with a $p < 0.001$ and these data are shown in *S6 Table*.

## Transfection with HIV-1 vectors

HIV proviral fragments, designated regions 1–17 (*S3 Table*), were cloned in vector pGL4.23 (Addgene, Cat: E8411) upstream of the luciferase reporter gene. pcDNA3.1+ (ThermoFisher, Cat: V79020) was used as a negative control. The pNL4-3-△5' construct was derived from pNL4-3 (GenBank: AF324493.2) using enzymatic digestion with *StuI* which cuts within the 5' cellular genomic sequence of the NL4-3 plasmid and within the *env* gene at 6,825 bp. Following digestion, pNL4-3-△5' was generated by blunt end ligation to delete the 5' end of the proviral genome. Successful deletion of 5' end was confirmed by DNA sequencing.

Transfection of HEK293T cells was performed in black-walled, clear-bottom 96-well plates. Cells were plated at a density of 20,000 cells/well 24 h prior to transfection. Cells were transfected at cell confluency of 60–70%. 300 ng of PEI and 50 ng of luciferase reporter constructs in up to 10 µl of Opti-MEM were transfected in triplicate wells. 24 hours post transfection, growth media was removed, and cells were lysed with cell culture lysis buffer for 10 min at room temperature. Firefly D-Luciferin was added at a 1:1 volume ratio and the plate was immediately read on BioTek Synergy HT plate reader at 260 nm for 1 sec/well.

## Western blot

Cells transfected with NL-43 or pNL4-3-△5' were lysed using freshly made lysis buffer (0.5% Igepal CA-630, Sigma; 50 mM Tris, pH 8.0; 150 mM NaCl; and Complete Protease Inhibitor (Millipore Sigma) in Milli-Q $H_2O$. Lysates were mixed in Laemmli's SDS-Sample Buffer (Boston BioProducts, Cat: BP-111R) and heated at 95°C for 5 min. Proteins were separated by 12% SDS-polyacrylamide gel electrophoresis (PAGE) and transferred by electroblotting onto polyvinylidene difluoride membrane (Millipore). After blocking 1 h with 5% nonfat dry milk, blots were incubated with primary antibodies (Monoclonal Mouse anti-β-Actin, BioRad, Cat: VMA00048 or Polyclonal Human Anti-HIV IgG, NIH-ARP3957, AIDS Reagent Program) overnight at 4°C and probed with horseradish peroxidase (HRP)-conjugated secondary antibodies. Membranes were developed with ECL Prime Western Blotting System (GE Healthcare) and visualized by merging a calorimetric (white light) and chemiluminescent image.

## Statistical analysis

All *in vitro* experiments and technical replicates were conducted at least three times. Experiments with primary cells included cells from at least three different donors. Data are presented as mean values ± standard error of the mean. *p* values were calculated based on the two tailed two sample T-test using GraphPad Prism software unless otherwise noted. $^*$ denotes $p < 0.05$; $^{**}$ denotes $p < 0.01$ and $^{***}$ denotes $p < 0.001$.

## Supporting information

**S1 Table. Probe and Primer Sequences.**
(PDF)

**S2 Table. Participant Details.**
(PDF)

**S3 Table. 5' RACE Sequences.**
(XLSX)

**S4 Table. HIV Tiling Sequences.**
(XLSX)

**S5 Table. Transcription Factor Binding.**
(PDF)

**S6 Table. HIV-1 Baits 14+15 Motif Analysis.**
(XLSX)

**S1 Fig. IPDA and MDM infection time course.** (A) Schematic of primer (arrows) and probe (stars) binding sites used for IPDA. For details see methods. (B) IPDA time course data for HIV-1 NL4-3-BaL infected MDMs. MDMs differentiated from 3 separate donors were infected as described in Methods and incubated for either 2 or 6 days before DNA isolation. IPDA was used to estimate intact and defective provirus frequencies per $1 \times 10^6$ cells at each time point. Day 2 and Day 6 intact and defective provirus estimates were not statistically different when analyzed by Two-Sample T Test.
(TIF)

**S2 Fig. Reverse transcriptase inhibitor Efavirenz (EFV) treatment of CD4+ T cells and MDMs prevented intact and defective HIV provirus establishment regardless of activation state.** (A) IPDA data for EFV pretreated CD4+ T cells infected with HIV-1NL4-3. For CD4+ T cells, EFV was added at a concentration of 10 μM 30 minutes prior to spinoculation with HIV-1NL4-3. Infection was limited to a single round by addition of the viral protease inhibitor Saquinavir 30 minutes after spinoculation. Resting and activated CD4+ T cells were participant matched. Data are from 3 separate donors. (B) IPDA data for EFV pretreated MDMs infected with HIV-1NL4-3-BaL. For MDMs, EFV was added at a concentration of 10 μM 24-hours prior to addition of HIV-1NL4-3-BaL to the cell cultures. Data are from 3 separate donors.
(TIF)

**S3 Fig. Multiplex RT-ddPCR probe locations and representative data.** (A) Schematic of RT-ddPCR probe binding sites aligned to the HIV genome. Traditional HIV transcripts are labeled with population numbers that correlate to the droplet populations shown in (B). Putative cryptic HIV transcripts lacking 5'UTR sequence are labeled as populations 2, 6, and 5. Population 1 represents empty droplets or HIV transcripts which lack all probed sequence sites. (B) Representative data acquired during multiplex RT-ddPCR. Data shown reflects the following multiplexed probe reactions: 1X concentration LTR probe reaction (y-axis, FAM), 0.5X Nef probe reaction (y-axis, FAM), and 1X Env probe reaction (x-axis, VIC). The multiplexed assay detected 8 distinct populations which were manually gated as reflected by different droplet colors and numbers above. Numbers represent droplets harboring the following distinct transcriptional species: (1) Empty droplets, (2) Nef Only Transcripts, (3) LTR Only Transcripts, (4) LTR+Nef Transcripts, (5) Env Only Transcripts, (6) Env+Nef Transcripts, (7) LTR+Env

Transcripts, (8) LTR+Env+Nef Transcripts.
(TIF)

**S4 Fig. Env+Nef HIV transcripts are polyadenylated.** RT was performed as described in Methods using a poly d(T) primer for reverse transcription of 500ng of RNA from either HIV-$1_{NL4-3-BaL}$ infected MDMs or HIV-$1_{NL4-3}$ infected CD4+ T cells which were unstimulated or activated for 72 hours using anti-CD3/CD28 beads. No Infection negative control sample is representative of N = 1. All remaining data are representative of three separate infections. CD4 + T cell data are donor matched between unstimulated and activated cells.
(TIF)

## Acknowledgments

We thank the Providence/Boston CFAR Basic Science and Biobehavioral Cores (P30 AI042853) and Dr. Nelson Lau of the BUSM Genome Science Institute for assistance in droplet digital PCR protocols. We also thank Allison Burdie who provided technical support in characterizing putative promoters and Drs. Manish Sagar and Suryaram Gummuluru (Boston University School of Medicine) for critical comments regarding the manuscript.

## Author Contributions

**Conceptualization:** Jeffrey Kuniholm, Elise Armstrong, Juan I. Fuxman Bass, Andrew J. Henderson.

**Data curation:** Jeffrey Kuniholm, Elise Armstrong, Brandy Bernabe, Carolyn Coote, Anna Berenson, Samantha D. Patalano, Alex Olson, Xianbao He, Juan I. Fuxman Bass.

**Formal analysis:** Jeffrey Kuniholm, Elise Armstrong, Brandy Bernabe, Carolyn Coote, Anna Berenson, Samantha D. Patalano, Alex Olson, Xianbao He, Nina H. Lin, Juan I. Fuxman Bass, Andrew J. Henderson.

**Funding acquisition:** Juan I. Fuxman Bass, Andrew J. Henderson.

**Investigation:** Jeffrey Kuniholm, Elise Armstrong, Brandy Bernabe, Carolyn Coote, Nina H. Lin, Juan I. Fuxman Bass.

**Methodology:** Jeffrey Kuniholm, Elise Armstrong, Brandy Bernabe, Carolyn Coote, Anna Berenson, Samantha D. Patalano, Alex Olson, Xianbao He, Juan I. Fuxman Bass.

**Resources:** Alex Olson, Nina H. Lin, Juan I. Fuxman Bass, Andrew J. Henderson.

**Supervision:** Juan I. Fuxman Bass, Andrew J. Henderson.

**Validation:** Jeffrey Kuniholm.

**Writing – original draft:** Jeffrey Kuniholm, Elise Armstrong, Andrew J. Henderson.

**Writing – review & editing:** Jeffrey Kuniholm, Alex Olson, Juan I. Fuxman Bass, Andrew J. Henderson.

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
