## [Decision Letter · Decision Letter 0]

29 Oct 2021

Dear Dr Henderson,

Thank you very much for submitting your manuscript "Intragenic proviral elements support transcription of defective HIV-1 proviruses" for consideration at PLOS Pathogens. As with all papers reviewed by the journal, your manuscript was reviewed by members of the editorial board and by several independent reviewers. In light of the reviews (below this email), we would like to invite the resubmission of a significantly-revised version that takes into account the reviewers' comments.

Reviewers 1 and 2 make specific requests that you should address experimentally to the greatest extent possible.  Reviewer 3 raises some broader issues that you should address as best you can.

We cannot make any decision about publication until we have seen the revised manuscript and your response to the reviewers' comments. Your revised manuscript is also likely to be sent to reviewers for further evaluation.

Sincerely,

Ronald Swanstrom

Associate Editor

PLOS Pathogens

Alexandra Trkola

Section Editor

PLOS Pathogens

Kasturi Haldar

Editor-in-Chief

PLOS Pathogens

orcid.org/0000-0001-5065-158X

Michael Malim

Editor-in-Chief

PLOS Pathogens

orcid.org/0000-0002-7699-2064

Reviewer's Responses to Questions

**Part I - Summary**

Reviewer #1: This manuscript from Kuniholm and colleagues describes a study of the transcriptional potential of defective HIV-1 proviruses. Infections were performed in resting and activated CD4+ T cells and macrophages. Using an IPDA, the investigators observed that resting CD4+ T cells and macrophage infections resulted in a majority of proviral integrants being defective. In contrast, the vast majority of integrants in activated CD4+ T cells appeared to be intact. The analysis of viral transcripts from these defective proviruses identified viral transcripts that lacked a 5’ untranslated UTR, suggesting that these transcripts did not arise from the viral 5’ LTR. Additionally, 5’ UTR deficient transcripts were identified in patients on suppressive antiviral therapy. Luciferase reporter plasmids were constructed in which various HIV-1 genomic sequences could serve as the promoter. Results from these reporters suggested that transcription can be directed by viral sequences between pol and env, or within env coding sequences. Finally, a NL4-3 proviral construct with deletion of sequences 5’ to env was found to express RNA and perhaps 35 and 17 kDa proteins.

This is a well-written and clearly presented manuscript that provides insight into the ability of defective HIV-1 proviruses to generate viral transcripts and perhaps proteins. As discussed in the manuscript, these viral molecules may contribute to chronic immune activation in patients on suppressive ART. Therefore, the manuscript provides a mechansims involved in chronic immune activation. Additionally, although there are hints in the literature that internal sequences in the provirus may have promoter activity, this study provides some of the best data on that point and confirms that aberrant viral transcripts occur in patient samples.

Minor points:

1. It may be relatively straightforward to identify the transcriptional start site from the Luciferase reporter plasmids (#14, 15) and the key TF binding sites. The inclusion of these data would strengthen the study.

2. On line 234, the Authors use the term “pauciclonal” to describe the progressive accumulation of defective proviral genomes. If pauciclonal means rare clonal proviruses, isn’t its use rather contradictory.

Reviewer #2: In this manuscript by Kuniholm et al., the authors described a novel cis-regulatory element in the HIV genome: that a segment within the HIV env genome may serve as an alternative promoter followed by a transcription start site that drive 3’ HIV transcription. This finding by itself is novel. There are a few interpretations that should be made more prudent/clear to ensure that the statements are well justified.

Reviewer #3: Kuniholm and colleagues report evidence that intragenic elements in the HIV genome support transcription of aberrant HIV-1 RNAs from defective proviruses that lack typical LTR promoter sequences. Much of the data is generated in cells following HIV infection in vitro. Using IPDA they find that resting CD4+ T cells and MDMs contain a larger proportion of defective proviruses. They measure Env and Nef transcripts generated outside of the HIV LTR, and map potential start sites in Env. They hypothesize that such transcripts may persistently expressed to potentially drive chronic inflammation in treated HIV+ people.

The work provides mechanistic support for the widely discussed hypothesis, following the description of “non-expressed” but transcriptionally active proviruses, that proposes a pathogenic role for the potential translational products of these viral RNAs. However, the work could do more to quantify the frequency and extent of these products in vivo

**Part II – Major Issues: Key Experiments Required for Acceptance**

Reviewer #1: (No Response)

Reviewer #2: 1. Figure 1: The authors interpret the lack of 5’ ddPCR probe detection as “the lack of HIV 5’ LTR”. This is wrong. The absence of the 5’ ddPCR signal only indicates that one of the 3 components of the ddPCR – the forward primer, the reverse primer, or the probe – does not bind. The biological context is that if the 5’ LTR is absent, integrase will not be able to catch the 5’ LTR end for integration. What actually happened in these ddPCR is that the psi packaging signal (note: it is downstream of 5’ LTR and upstream of gag start codon; it does not have promoter function; it is downstream of transcription start site; and it should not be called as 5’ LTR) is deleted or having mismatches. Packaging signal deletion has been reported in many previous publication. Much of the LTR promoter function are driven by U3 (not R or U5), which is not covered by the Yukl R-U5-gag PCR. , Please remove all statements regarding such interpretations, such as “Proviruses lacking 5’ LTR are transcribed” or line 161 “defective proviruses lacking 5’ LTR….”.

2. Figure 4: The 5’ RACE PCR is the most important experiment in this manuscript, providing evidence that the HIV-1 3’ reads do not start with the canonical transcription start site (+1). The authors need to provide detailed sequences, aligned to NL4-3 reference, regarding where these reads are, such as Takata/Bieniesz/PLoS Pathogens 2018 Figure 9AB or Pollack/Ho Cell Host Microbe Figure 2. Please label potential TATA box and estimated transcription start site on such plots. Also, if the prediction is right, where is the first start codon? Does that generate an in-frame protein?

3. Figure 5: StuI has only one cut in the NL4-3 plasmid (one cut in env but not in LTR). Supposedly, cutting pNL4-3 with StuI and ligate back creates either just the original NL43 plasmid. How do the authors remove the “6,825 bp” genome in pNL43, if there is not another cut 6,825 bp upstream? If there is such a cut site, please draw the nucleotide sequence and annotate the cut site within LTR or the plasmid.

4. Table S4: Please list the sequence (transcription factor binding motif) of these candidate transcription factor binding site and make this into a figure. See examples: Figure 5B, Duverger/Kutsch/Journal of Virology 2013.

Reviewer #3: 1. CD4+ T cells or MDMs were either unstimulated or activated with anti-CD3/anti-CD28 beads prior to infection with HIV-1 VSVgNL4-3 or HIV-1NL4-3 BaL. It would be expected that different proportions of infection events in these cells under these conditions would create integrants and that differing proportions of these cells might survive in culture over time. Infected cells were cultured for at least 48-hours to allow the completion of reverse transcription and integration (Supplemental Figure 1B). Genomic DNA from infected cells was used as template for IPDA. This showed that activated CD4+ T cells had a higher frequency (>90%) of intact HIV provirus when compared to resting CD4+ T cells or MDMs (Figure 1A, 1B) which had on average 25% and 40% intact proviruses, respectively. Approximately 50-60% of proviruses detected in both resting CD4+ T cells and MDMs were defective with the majority of proviruses harboring 5’-deletions. However in vivo a substantial proportion of these activated cells may not be long-lived. Although this is a highly artificial (but useful) in vitro system, it would be very useful to perform serial studies of the cell populations over time, to ascertain if the predominant intact proviral is found largely in cells fated for short half-lives.

2. It would be very useful to directly enumerate how many aberrant transcripts are present per million CD4+ or MDM cells, and at least to provide some sort of measure of the distribution of these transcripts (ie millions in a few cells, a few in millions of cells).

3. AHI in vitro generates more intact proviruses, but these seem unlikely to persist, as above. Unstimulated T cells and MDM infection is likely to proceed more slowly in vitro and in vivo. It would be important to study HIV+ blood cells prior to ART (if samples are not available, they could be requested from clinical repositories, such as that of the ACTG). IPDA should be assessed PBMCs, sorted if possible in to T cell vs MDM populations, in the presence or absence of activation markers if sufficient cells can be obtained for study. Additional studies such as an attempt to correlate the transcripts expressed (Env, Nef) with the frequency of persistent immune responses (which may be already available from some samples in some prior studies) would strengthen the work.

**Part III – Minor Issues: Editorial and Data Presentation Modifications**

Reviewer #1: Minor points:

1. It may be relatively straightforward to identify the transcriptional start site from the Luciferase reporter plasmids (#14, 15) and the key TF binding sites. The inclusion of these data would strengthen the study.

2. On line 234, the Authors use the term “pauciclonal” to describe the progressive accumulation of defective proviral genomes. If pauciclonal means rare clonal proviruses, isn’t its use rather contradictory.

Reviewer #2: 1. Figure 2 title should be “CD4+ T cells and MDMs express aberrant HIV-1 RNAs…” Please add HIV-1 before RNA. The current title reads as CD4+ T cells are expressing aberrant cellular RNAs.

2. When describing HIV RNA, the authors should use italic, all lower case, such as env.

3. Figure 5: The HIV sequence annotations need to made in frame, such as those depicted by Los Alamos HIV Database. Please do not put gag (reading frame 1) and env (reading frame 3) together (as in Figure 5, Figure S1).

Figure S1 is wrong, as gag-pol are overlapping in different reading frames.

Figure S3 is wrong – as the viral genes are in 5 different reading frames (there should be only three)

4. Restriction enzyme name: the first 3 letters should be italic (Stu), denoting the species of bacteria where this enzyme derived from. The number (I) should be regular (non-italic).

5. Please avoid using the term “Patient”, per request by people living with HIV. Authors used the term people living with HIV and add Patient on top of them (Table S2 title) and no longer meet the purpose for this plead. The authors can use “participant matched” as opposed to “patient matched”.

6. HIV copy numbers need to be plotted in log scale to represent the exponential biological nature of HIV replication and cellular proliferation. Authors did comply with this, but please correct the following panels: Figure 5B, S1B, S2.

7. Table outlines in Table S1 are inconsistent and missing in some columns.

Reviewer #3: Is Line 75, Ref. 20 incorrectly placed? It does not seem directly relevant to the points made.

PLOS authors have the option to publish the peer review history of their article (what does this mean?). If published, this will include your full peer review and any attached files.

Reviewer #1: No

Reviewer #2: No

Reviewer #3: No
---

## [Editor Report · Decision Letter 1]

7 Dec 2021

Dear Dr Henderson,

We are pleased to inform you that your manuscript 'Intragenic proviral elements support transcription of defective HIV-1 proviruses' has been provisionally accepted for publication in PLOS Pathogens.

Best regards,

Ronald Swanstrom

Associate Editor

PLOS Pathogens

Alexandra Trkola

Section Editor

PLOS Pathogens

Kasturi Haldar

Editor-in-Chief

PLOS Pathogens

orcid.org/0000-0001-5065-158X

Michael Malim

Editor-in-Chief

PLOS Pathogens

orcid.org/0000-0002-7699-2064
---

## [Editor Report · Acceptance letter]

16 Dec 2021

Dear Dr Henderson,

We are delighted to inform you that your manuscript, "Intragenic proviral elements support transcription of defective HIV-1 proviruses," has been formally accepted for publication in PLOS Pathogens.

Best regards,

Kasturi Haldar

Editor-in-Chief

PLOS Pathogens

orcid.org/0000-0001-5065-158X

Michael Malim

Editor-in-Chief

PLOS Pathogens

orcid.org/0000-0002-7699-2064